# Lower infant mortality, higher household size, and more access to contraception reduce fertility in low- and middle-income nations

Corey J. A. Bradshaw[1,2☯], Claire Perry [3☯], Melinda A. Judge [4☯], Chitra M. Saraswati[4], Jane Heyworth[3], Peter N. Le Souëf[2,4☯] *

1 Global Ecology *Partuyarta Ngadluku Wardli Kuu*, College of Science and Engineering, Flinders University, Adelaide, SA, Australia, 2 National Health and Medical Research Council Special Initiative in Human Health and Environmental Change, Healthy Environments And Lives (HEAL) Network, Canberra, ACT, Australia, 3 School of Population and Global Health, The University of Western Australia, Crawley, WA, Australia, 4 School of Medicine, The University of Western Australia, Crawley, WA, Australia

☯ These authors contributed equally to this work.
* peter.lesouef@uwa.edu.au

**Data Availability Statement:** Demographic and Health Surveys (dhsprogram.com) and Multiple Indicator Cluster Surveys (mics.unicef.org) data

## Abstract

Although average contraceptive use has increased globally in recent decades, an estimated 222 million (26%) of women of child-bearing age worldwide face an unmet need for family planning—defined as a discrepancy between fertility preferences and contraception practice, or failing to translate desires to avoid pregnancy into preventative behaviours and practices. While many studies have reported relationships between availability/quality of contraception and family planning, infant mortality, and fertility, these relationships have not been evaluated quantitatively across a broad range of low- and middle-income countries. Using publicly available data from 64 low- and middle-income countries, we collated test and control variables in six themes: (*i*) availability of family planning, (*ii*) quality of family planning, (*iii*) female education, (*iv*) religion, (*v*) mortality, and (*vi*) socio-economic conditions. We predicted that higher nation-level availability/quality of family-planning services and female education reduce average fertility, whereas higher infant mortality, greater household size (a proxy for population density), and religious adherence increase it. Given the sample size, we first constructed general linear models to test for relationships between fertility and the variables from each theme, from which we retained those with the highest explanatory power within a final general linear model set to determine the partial correlation of dominant test variables. We also applied boosted regression trees, generalised least-squares models, and generalised linear mixed-effects models to account for non-linearity and spatial autocorrelation. On average among all countries, we found the strongest associations between fertility and infant mortality, household size, and access to any form of contraception. Higher infant mortality and household size increased fertility, whereas greater access to any form of contraception decreased fertility. Female education, home visitations by health workers, quality of family planning, and religious adherence all had weak, if any, explanatory power. Our models suggest that decreasing infant mortality, ensuring sufficient housing to reduce household size, and increasing access to contraception will have the greatest effect on decreasing global fertility. We thus provide new evidence that progressing

from the following countries were used (3-letter ISO): Latin America & Caribbean (COL, DOM, GTM, HTI, HND, PER); Sub-Saharan Africa (AGO, BEN, BFA, BDI, CMR, TCD, COM, COD, COG, CIV, GNQ, ETH, GAB, GMB, GHA, GIN, KEN, LSO, LBR, MDG, MWI, MLI, MRT, MOZ, NAM, NER, NGA, RWA, STP, SEN, SLE, ZAF, TZA, TGO, UGA, ZMB, ZWE); Middle East & North Africa (EGY, JOR, YEM); Europe & Central Asia (ALB, ARM, KGZ, TJK, TUR); South Asia (AFG, BGD, IND, MDV, NPL, PAK), East Asia & Pacific (KHM, IDN, LAO, MMR, PNG, PHL, TLS). Also incorporated was data from the Family Planning Effort database (2017 version) (track20. org/pages/data_analysis/policy/FPE.php), National Composite Index on Family Planning (track20.org/ pages/data_analysis/policy/NCIFP.php), World Bank (data.worldbank.org), Association of Religion Data Archives (thearda.com), the World Inequality Database (wid.world/data) and World Health Organization Global Health Observatory data repositories (who.int/data/gho). A detailed description of the data used is included in the supplementary information, section 2. Supplementary figure 1 was created using the freely available program QGIS Version 3.26.1-Buenos Aires with an open-licence layer (https:// hub.arcgis.com/datasets/esri::world-countries-generalized/explore?location=-0.684955%2C0. 000000%2C2.40).

**Funding:** CB was supported in part by a Rockefeller Foundation Bellagio Writer's Fellowship. The authors received no other specific funding for this work.

the United Nation's Sustainable Development Goals for reducing infant mortality can be accelerated by increasing access to family planning.

## Introduction

Although average contraceptive use has increased globally in recent decades, an estimated 222 million (26%) women of child-bearing age worldwide face an unmet need for family planning —defined as a discrepancy between fertility preferences and contraception practice, or failing to translate desires to avoid pregnancy into preventative behaviours and practices [1]. The United Nation's Sustainable Development Goals 3 and 5 emphasise the basic right to exercise control over sexual and reproductive health through universal access to family planning [2]. While achieving Goal 3 is targeted for 2030, reducing global maternal mortality to $< 70$ per 100,000 live births and under-5 mortality to $\leq 25$ per 1,000 live births are not on track to be met [3]. Providing readily available, high-quality family-planning services is necessary because this is expected to decrease not only fertility, but also the number of unintended pregnancies and infant and maternal deaths [4,5]. Allowing individuals to be able to decide to have fewer children also has the potential to facilitate better investment in the overall health and well-being of families and communities [5].

To date, there is no uniform measure (i.e., set of indicators) for availability and quality of family planning; therefore, gauging the effects these might have on fertility is difficult. Few studies have investigated the relationship between socio-economic conditions and fertility among nations; there are also few studies on the availability and quality of family planning that do more than merely hypothesise a generally negative association with fertility [6,7]. We investigated the association between fertility and the availability and quality of family planning, as well as the potential effects of education, household size (a proxy for population density), religion, infant mortality, and socio-economic conditions identified from the literature. For example, higher maternal education is correlated with reduced total fertility [8]. Religious adherence can also affect fertility patterns [9]—e.g., Catholicism expressly forbids contraception, and predominantly Muslim countries in Sub-Saharan Africa tend to have higher total fertility rates than non-Muslim countries [10]—although the relationship is complex and not always present [11,12]. Infant mortality has been linked to increased fertility due to the 'insurance' or 'replacement' effect [13], and child morbidity and mortality are themselves exacerbated by higher household sizes (i.e., number of individuals per household) [14–20].

To test the complex potential relationships between these and other hypothesised drivers of human fertility, we specifically tested whether: (1) increasing the availability of family planning is associated with reduced fertility; (2) increasing the quality of family-planning services is associated with reduced fertility; (3) increasing years of female education is associated with reducing fertility; (4) increased fertility is observed in countries with a higher prevalence of adherents of Catholicism or Islam; (5) higher mean household size is positively correlated with fertility, and (6) lower socio-economic conditions and (7) higher mortality (infant and maternal) are associated with higher fertility.

## Methods

### Data

We used aggregated data at the national level, collating publicly available data from several online sources. These included Demographic and Health Surveys [21], Family Planning Effort

Index [22], Multiple Indicator Cluster Surveys [23], National Composite Index on Family Planning [24], the World Bank [25], the World Inequality Database [26], the Association of Religion Data Archives [26,27], and the World Health Organization Global Health Observatory data repositories [28]. We obtained most of the required data from the Demographic and Health Surveys; these nationally representative surveys had uniform methods and included a similar period of reference when measuring indicators.

From these datasets, we derived the following test variables for model construction (additional information in Supporting information Section 2). As the response, we used *fertility*, which is the mean number of children a woman has between the ages of 15 to 49 years [29], which we sourced from the World Bank (mean from 2010–2020) [25]. We separated the modelling into to two main phases (see *Analyses*) that first tested relationships to fertility within six separate themes: (*i*) availability of family planning, (*ii*) quality of family planning, (*iii*) female education, (*iv*) religion, (*v*) mortality, and (*vi*) socio-economic conditions. The second phase incorporated the top-ranked indicators into a final model set.

For **theme i**, *availability of family-planning services* is measured via the 'access' index from the Family Planning Effort Index (2014 version) [22]. We added other indicators from this dataset that measured aspects of family planning-service delivery, including access to community health workers (extent of population visited by healthcare workers who educate about family planning and maternal and child health). For **theme ii**, we derived *quality of family-planning services* using the 'quality' index from the National Composite Index on Family Planning [24] (2014 version). We also included additional indicators from the Demographic and Health Surveys and Multiple Indicator Cluster Surveys [23] (most recent data per country) to measure other aspects not included in the 'quality' index from the National Composite Index on Family Planning database [24]. For **theme iii**, we obtained *mean years of education of women aged 15–49 (weighted)* from the Demographic and Health Surveys and Multiple Indicator Cluster Surveys [23]. For **theme iv** (*religion*), we combined the percentage of the population adhering either to Catholicism or Islam from the Association of Religion Data Archives [26]. For **theme v**, infant mortality is higher in areas of low socio-economics [30,31] and a known correlate with fertility [32], so we included the most recent infant mortality data (deaths per 1000 live births) for each country from the World Health Organization Global Health Observatory data repository [28]. We also included the number of maternal deaths per 100,000 live births, and the average number of battle-related deaths (2015–2020) per capita (most recent data from the World Bank) [25]. For **theme vi** (*socio-economics*), women in the lowest quintile of household wealth have less availability of family planning compared with women from higher-wealth households [33]. We therefore obtained (*a*) the net personal wealth of the bottom 50% of the population (*p0p50*) from the World Inequality Database [26]. We also included the (*b*) mean number of household members as a proxy for population density, and (*c*) percentage of households with three generations residing (from the Demographic and Health Surveys and Multiple Indicator Cluster Surveys) [23] (each country's most recent data).

## Analyses

**Data preparation.**    We applied descriptive analyses to provide an overview of the distribution across all variables. We first transformed all variables and scaled/centred them to improve homoscedasticity and Gaussian behaviour using a logit transformation of the base proportion and then centring and scaling using the *scale* function in the R programming language [34]. In the case of non-proportional variables (*female years of education*, *battle-related deaths*, *maternal mortality*, *net personal wealth in the bottom 50%*), we either scaled the raw values (*female*

*years of education*, *battle-related deaths*, *maternal mortality*), or scaled the cube of the variable (*net personal wealth*) to give approximate-Gaussian behaviour. We examined the transformed explanatory variables for collinearity using a non-parametric (Kendall's $\tau$) correlation matrix (S1 Table).

**Multiple imputation by chained equations.** The final dataset included 64 countries with relevant data, although there were some missing values per variable (missing: 19% *visitation by community health workers*; 6% *female years of education*; 9% *net personal wealth bottom 50%*; 9% *quality of family planning index*; 5% *access to any form of contraception*). Removing these countries from analysis would have reduced our sample from 64 to 51 countries, a large reduction considering the complexity of the hypotheses to test. We therefore ran multiple imputation by chained equations using the mice library in R [35] to impute any missing values in the predictor variables based on the final-phase dataset (see below) to maintain the highest-possible sample size of 64 countries for subsequent analysis. Multiple imputation using this method is robust for up to 75% missing values [36], and provides stronger inferences than ignoring missing data [37]. Assuming values were missing at random, we employed predictive means matching in the *mice* function with 50 maximum iterations to impute the missing data. Results were similar whether using imputed or missing-values datasets (see Results).

**Linear models.** We built general linear models with the *glm* function in R to identify the contributory (transformed) variable with the most explanatory power in each of the six themes (see Supporting information Sections 4–7). We included various models in each theme and ranked them based on the Bayesian information criterion (BIC) [38,39] given that our focus was on identifying the main drivers of variance in fertility as opposed to prediction [40], with relative model probability equal to its BIC weight (*w*BIC) (S2–S5 Tables, Supporting information Sections 4–7).

We suspected potential spatial autocorrelation among the country values, so we also constructed general linear mixed-effects models using the lme4 package [41] in R, coding a random effect according to major world region (S1 Fig). Here, we aggregated the 64 countries into the six World Bank regions—Latin America & Caribbean (*n* = 6 countries); Sub-Saharan Africa (*n* = 37); Middle East & North Africa (*n* = 3); Europe & Central Asia (*n* = 5), South Asia (*n* = 6); East Asia & Pacific (*n* = 7)—to account partially for spatial autocorrelation (see Supporting information Section 1, S1 Fig and *Analyses*). Including the top-ranked variables from each theme into a final model set, we determined both the evidence for a non-random effect of the final variables on fertility, as well as the goodness of fit (percent deviance explained per model).

**Boosted regression trees.** We also built boosted-regression trees [42] of the final model to account for potential nonlinearity in the relationships between the fertility response and the potential indicators (S2–S5 Figs, Supporting information Sections 4–7). Mixed-effects models developed in the previous section potentially miss sub-regional spatial autocorrelation, so to account for a deeper level of spatial autocorrelation and to quantify uncertainty in the relationships between fertility and each explanatory variable, we resampled countries in the dataset with replacement 1000 times. We then passed each resampled dataset to the boosted regression trees and then calculated the 2.5th and 97.5th percentiles for the respective distribution for each predicted fertility as the uncertainty bounds. We applied kappa ($\kappa$) limitation to the resampled selections to limit the influence of outliers [43], where we retained only the resampled mean ranks within $\kappa\sigma$ of the overall average mean ($\kappa$ = 2). We then recalculated the average and standard deviation of the mean rank, with the process repeated five times.

**Generalised least-squares models.** Finally, we applied general least-squares models that are designed explicitly to account for spatial autocorrelation among spatial units (countries, in this case) to the final model set. For each country, we coded the centroid coordinates (in

latitude/longitude) and determined that a Gaussian correlation was the top-ranked, within-group correlation structure for the saturated model (although the spherical correlation structure was nearly identically supported); we therefore ran the models in the final phase as per the general linear/mixed-effects models with a Gaussian correlation structure. We ranked the ensuing models according to $w$BIC, and calculated relative goodness-of-fit using three different pseudo-$R^2$ metrics: McFadden, Cox and Snell, and Craig and Uhler (using the *nagelkerke* function in R library rcompanion) [44]. All data and R code to repeat the analyses are provided at github.com/cjabradshaw/humanfertility (doi:10.5281/zenodo.7496142).

## Results

Fertility was Normally distributed (Shapiro-Wilk normality test: $W$ = 0.970; $p$ = 0.123) with one outlier (Niger), although that country's value of 7.14 per woman is the highest national fertility globally [25]. A non-parametric (Kendall's $\tau$) correlation matrix of the highest-ranked variable from each initial model (S1 Table) indicated that the strongest correlation observed was the relationship between access to any form of contraception and household size ($\tau$ = -0.516). Of all regions, countries in sub-Saharan Africa had the highest average number of infant mortalities (45 per 1000 live births), as well as the highest average fertility (4.8 per woman) (Fig 1).

### General linear and linear mixed-effects models

*Visitation by community health workers* (CHW) was the most supported variable in theme 1 (S2 Table, S2 Fig), *access to any type of contraception* (ACC) and *quality of family planning index* (QUA) in theme 2 (S3 Table, S3 Fig), *female years of education completion* (FE) in theme 3, *percentage Catholic or Muslim* in theme 4 (religion), *infant mortality* (IM) in theme 5 (S4 Table, S4 Fig A-4), and *household size* in theme 6 (HS) (S5 Table, S5 Fig). Based on this group of seven variables, we constructed 26 candidate general linear and general linear mixed-effects models (Tables 1 and 2, respectively) to determine the most-supported models according to $w$BIC.

According to the final-phase general linear models, the model with *infant mortality* and *access to any form of contraception* had the highest model support ($w$BIC = 0.71) and explained 62.9% of the deviance, followed by the saturated model ($w$BIC = 0.15; deviance explained = 71.8%) (Table 1). *Infant mortality* had the highest % deviance explained (49.7%) among the single-variable models (Table 1).

For the generalised linear mixed-effects models accounting for the random effect of region, the model including *female years of education* and *access to any form of contraception* ($w$BIC = 0.60), but the fixed effects accounted for over half (57%) of the total variance explained ($R_m$ = 43.6% of 76.6%; Table 2). *Female years of education* had the highest explanatory power ($R_m$ = 34.0%) of any single-variable model, followed by *mean number of individuals per household* (Table 2).

### Boosted regression trees

The boosted regression trees using either raw, untransformed values (grey bars in Fig 2A), or the resampled relative contributions across 1000 iterations of the transformed variables (black bars in Fig 2A), clearly indicated that *infant mortality* had the strongest explanatory power for variance in fertility among countries (Fig 2A), followed by *household size*, and *access to any form of contraception* (Fig 2A) (see also S2–S4 Figs, Supporting information Sections 4–8). The relationship between *infant mortality* and fertility was strongly stepped, suggesting a threshold of a sharp increase in fertility occurring once a country exceeded an infant mortality of ~

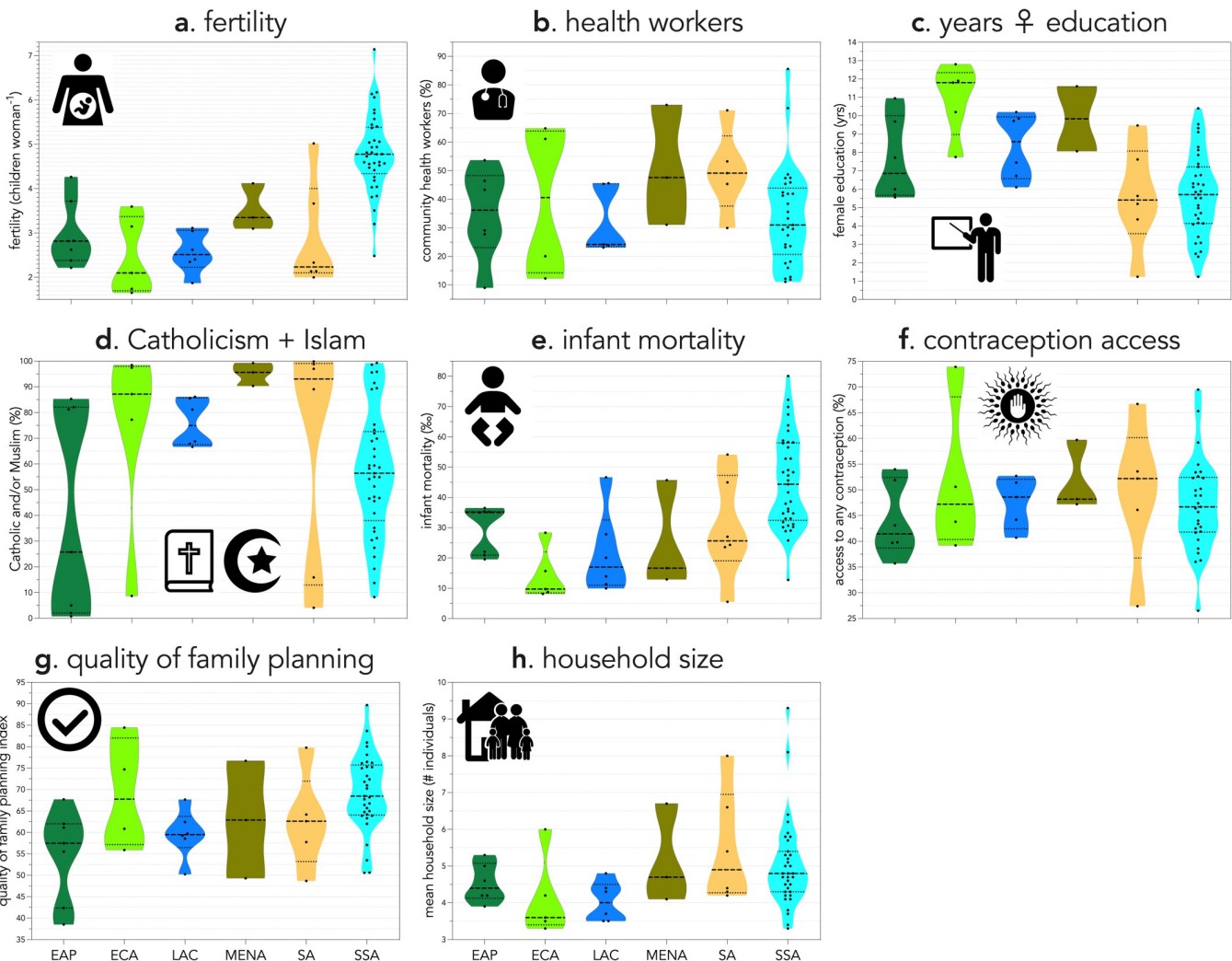

**Fig 1.** Violin plots of country-level raw values for **a**. fertility (children woman$^{-1}$) and the highest-ranked variables from each of the six initial modelling phases —**b**. population with access to community health workers (%), **c**. female years of education completed, **d**. population claiming adherence to Catholicism or Islam (%), **e**. infant mortality (per 1000 births), **f**. population with access to any form of contraception (%), **g**. quality of family planning index, **h**. mean number of household members. The 64 low- and middle-income countries with full data included in the analysis are classed into four world regions (EAP = East Asia and Pacific; ECA = Europe and Central Asia; LAC = Latin America and Caribbean; MENA = Middle East and North Africa; SA = South Asia; SSA = sub-Saharan Africa). Country codes per region (3-letter ISO): **EAP** (KHM, IDN, LAO, MMR, PNG, PHL, TLS); **ECA** (ALB, ARM KGZ, TJK, TUR); **LAC** (COL, DOM, GTM, HTI, HND, PER); **MENA** (EGY, JOR, YEM); **SA** (AFG, BGD, IND MDV, NPL, PAK); **SSA** (AGO, BEN, BFA, BDI, CMR, TCD, COM, COD, COG, CIV, GNQ, ETH, GAB, GMB, GHA, GIN, KEN, LSO, LBR, MDG, MWI, MLI, MRT, MOZ, NAM, NER, NGA, RWA, STP, SEN, SLE, ZAF, TZA, TGO, UGA, ZMB, ZWE).

0.0282 (Figs 2B and 3). There was a positive relationship between fertility and *household size*, with higher household sizes having higher fertility (Fig 2C). *Access to contraception* had a negative relationship with fertility, suggesting that fertility would decrease as a country improves its citizens' access to contraception (Fig 2D). Interestingly, the predicted relationship between *quality of family planning* was opposite to the expected direction: higher 'quality' was correlated with higher fertility (Fig 2E) (see Discussion for possible reasons). While the relationships between fertility and education, religion, and *health worker visitation* were in the hypothesised directions (Fig 2F–2H), their relative contributions were relatively weak (Fig 2A). The relative contribution rank of predictor variables was identical using either the imputed or restricted datasets (S6 Fig).

**Table 1. Candidate general linear models of the highest-ranked variable from each theme (1. availability of family planning, 2. quality of family planning, 3. education, 4. religion, 5. mortality, 6. socio-economics) in relation to variation in fertility among 64 low- and middle-income countries.**

| model | $k$[a] | LL[b] | ΔBIC[c] | $w$BIC[d] | %DE[e] |
|---|---|---|---|---|---|
| IM[f]+ACC[g] | 3 | -75.53 | 0.00 | 0.71 | 62.9 |
| *ALL*[h] | 8 | -66.72 | 3.16 | 0.15 | 71.8 |
| IM+HS[i] | 3 | -77.78 | 4.50 | 0.07 | 60.2 |
| FE[j]+IM | 3 | -78.34 | 5.61 | 0.04 | 59.5 |
| FE+ACC | 3 | -78.89 | 6.72 | 0.02 | 58.8 |
| IM+QUA[k] | 3 | -82.04 | 13.01 | <0.01 | 54.5 |
| CM[l]+IM | 3 | -82.34 | 13.61 | <0.01 | 54.1 |
| IM | 2 | -85.23 | 15.24 | <0.01 | 49.7 |
| ACC | 2 | -86.93 | 18.63 | <0.01 | 47.0 |
| CHW[m]+IM | 3 | -85.23 | 19.40 | <0.01 | 49.7 |
| FE+HS | 3 | -85.61 | 20.15 | <0.01 | 49.2 |
| FE | 2 | -87.92 | 20.61 | <0.01 | 45.4 |
| FE+QUA | 3 | -85.86 | 20.66 | <0.01 | 48.8 |
| CHW+ACC | 3 | -86.26 | 21.44 | <0.01 | 48.1 |
| CHW+FE | 3 | -86.91 | 22.75 | <0.01 | 47.0 |
| CM+ACC | 3 | -86.92 | 22.78 | <0.01 | 47.0 |
| HS | 2 | -93.66 | 32.09 | <0.01 | 34.6 |
| CHW+HS | 3 | -91.90 | 32.74 | <0.01 | 38.1 |
| CM+HS | 3 | -92.49 | 33.92 | <0.01 | 37.0 |
| *intercept-only* | 1 | -107.25 | 55.12 | <0.01 | - |
| CHW | 2 | -105.73 | 56.23 | <0.01 | 4.7 |
| CHW+QUA | 3 | -103.81 | 56.56 | <0.01 | 10.2 |
| QUA | 2 | -106.21 | 57.19 | <0.01 | 3.2 |
| CM | 2 | -106.31 | 57.39 | <0.01 | 2.9 |
| CHW+CM | 3 | -104.72 | 58.37 | <0.01 | 7.6 |
| CM+QUA | 3 | -104.91 | 58.76 | <0.01 | 7.1 |

[a]$k$ = number of parameters; [b]LL = log-likelihood; [c]ΔBIC = difference in Bayesian information criterion between model and top-ranked model; [d]$w$BIC = Bayesian information criterion weight (≈ model probability); [e]%DE = % deviance explained; [f]IM = infant mortality; [g]ACC = access to any form of contraception; [h]ALL = saturated model (including the six highest-ranked variables from each phase); [i]HS = mean number of individuals per household; [j]FE = female years of education completion; [k]QUA = quality of family planning index; [l]CM = % Catholic or Muslim; [m]CHW = visitation from community health workers.

### General least-squares models

The country centroids explained 7–14% of the variation in the general least-squares models. After accounting for spatial relationships, *female years of education*, *household size*, *access to any form of contraception*, and *infant mortality* emerged as the most important correlates to variation in fertility among countries (Table 3).

### Discussion

Our study is the first to investigate the potential associations between the availability and quality of family planning and fertility, while simultaneously considering other potential contributory variables among 64 low- and middle-income countries. We found that high infant mortality was most strongly related to high fertility, but nonlinearly, followed by high household size (i.e., mean number of individuals per household) and reduced access to contraception. However, the relative contribution of each of these variables to reducing fertility potentially also needs to consider the correlation between access to contraception and

**Table 2. Generalised linear mixed-effects models (only 10 top-ranked models according to $w$BIC$_c$ shown) of the highest-ranked variable from each theme (1. availability of family planning, 2. quality of family planning, 3. education, 4. religion, 5. socio-economics) in relation to variation in fertility among 46 low- and middle-income countries.**

| model | $k^a$ | LL$^b$ | $\Delta$BIC$^c$ | $w$BIC$^d$ | R$_m{}^e$ | R$_c{}^f$ |
|---|---|---|---|---|---|---|
| FE$^g$+ACC$^h$ | 9 | -62.34 | 0.00 | 0.60 | 43.6 | 76.6 |
| FE+HS$^i$ | 11 | -62.81 | 0.96 | 0.37 | 40.3 | 77.2 |
| IM$^j$+HS | 18 | -66.36 | 8.04 | 0.01 | 42.0 | 71.9 |
| FE | 20 | -68.54 | 8.25 | 0.01 | 34.0 | 71.1 |
| HS | 25 | -69.57 | 10.30 | <0.01 | 29.2 | 70.6 |
| FE+IM | 8 | -67.89 | 11.11 | <0.01 | 41.1 | 71.4 |
| ACC | 23 | -70.87 | 12.91 | <0.01 | 30.4 | 69.4 |
| IM+ACC | 16 | -68.81 | 12.94 | <0.01 | 42.1 | 69.9 |
| CHW$^k$+FE | 2 | -69.83 | 14.99 | <0.01 | 34.2 | 70.8 |
| FE+QUA$^m$ | 10 | -70.01 | 15.35 | <0.01 | 33.4 | 70.9 |
| . . . | | | | | | |

$^a k$ = number of parameters; $^b$LL = log-likelihood; $^c \Delta$BIC = difference in Bayesian information criterion between model and top-ranked model; $^d w$BIC = Bayesian information criterion weight ($\approx$ model probability); $^e R_m$ = marginal $R^2$; $^f R_c$ = conditional $R^2$; $^g$FE = female years of completed education; $^h$ACC = access to any form of contraception; $^i$HS = mean number of individuals per household; $^j$IM = infant mortality; $^k$CHW = visitation from community health workers; $^l$QUA = quality of family planning index.

decreased infant mortality. Increased access to contraception has been previously associated with decreased infant mortality and is thought to act through increasing space between births, and avoiding the increased infant mortality found with higher birth order [45,46]. We also found that other factors considered to be important in reducing fertility, including community health worker visits, female education, and religion (Catholicism or Islam) had only weak associations, at least at the spatial scale across nations.

The counter-intuitive relationship we found between the index of family-planning quality (increasing quality leading to higher fertility) might be explained in part by the dominance of nations in sub-Saharan Africa in our sample. A recent study [47] comparing contraceptive use and the quality index from the National Composite Index on Family Planning [24] indeed found a positive correlation between the two variables, yet the relationship was strongest for sub-Saharan Africa, despite this region having the lowest overall contraceptive use, highest fertilities, and highest infant mortality (Fig 1). In fact, sub-Saharan Africa scored highest across all five dimensions, as well as overall, on the National Composite Index on Family Planning, which might indicate biases that are a known limitation of the methodology used [47]. This potentially suggests that the combined metric of 'quality' could be interpreted differently in African nations relative to elsewhere.

Infant mortality as a strong predictor of variation in fertility is supported elsewhere [32]. Overall, our findings support the notion that to decrease global fertility, both infant survival rates plus access to contraception need to be increased. Recommendations for measures to decrease infant mortality emphasise improving the quality of antenatal care, increasing the number of trained healthcare staff at births, and improving postnatal care for both infants and mothers [48,49]. Given the evidence that household composition, including the number of individuals, can worsen child health [14–20], improving living conditions to ameliorate high-density living could also indirectly result in lower fertility. A greater emphasis on providing access to contraception as a direct contribution to decreasing infant mortality is important for such guidelines.

There are multiple factors that can influence fertility, some of the subtleties of which we could not test directly with the available data. Responsible family planning requires the effort

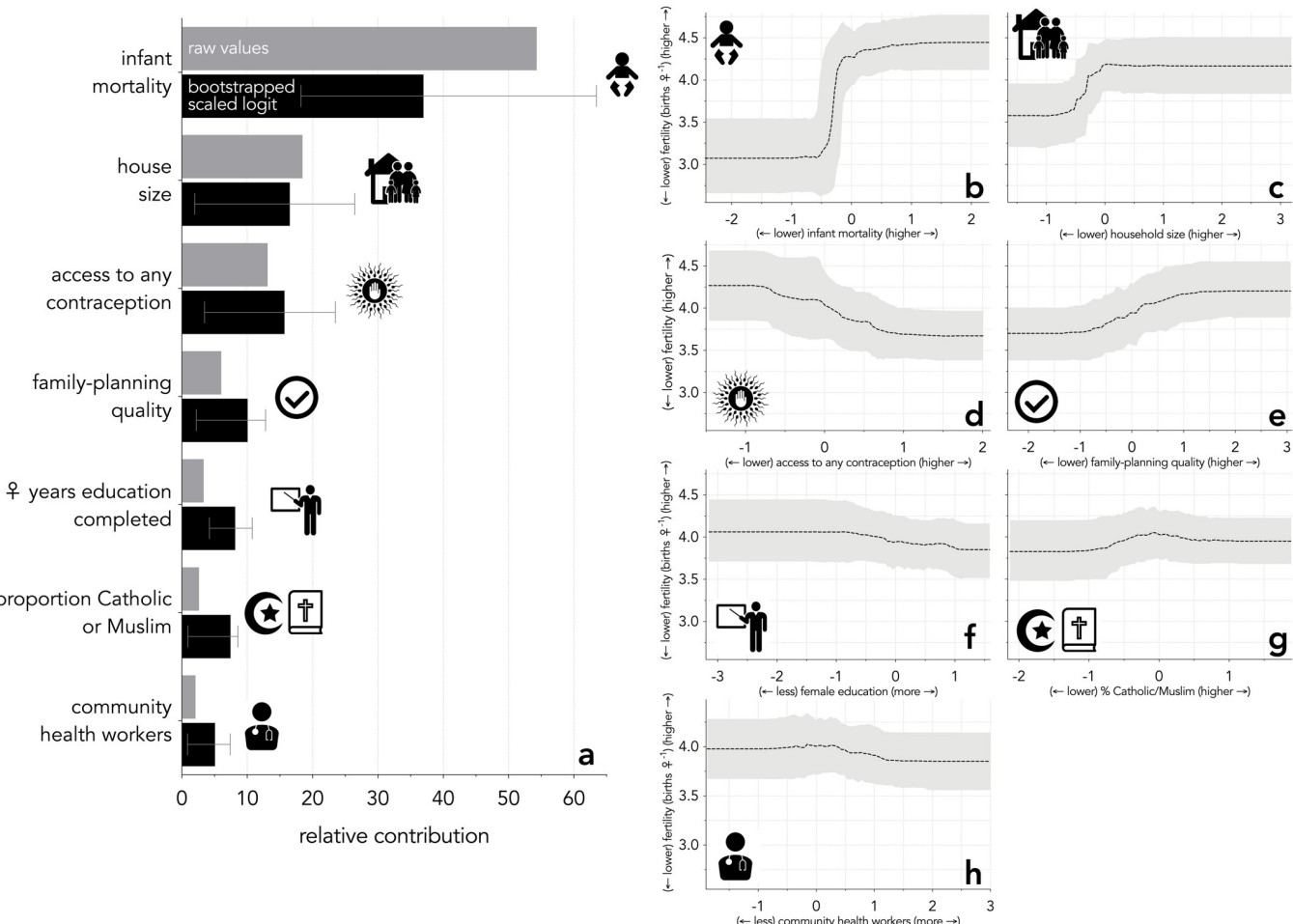

**Fig 2.** (**a**) The relative contribution and proportion of variance in indicators of family-planning availability, family-planning quality, education, religion, and socio-economics derived from boosted regression trees. The relative contribution of each variable (raw values) to the variance in fertility for each country in the dataset is represented by the grey bars. % deviance explained for raw and bootstrapped boosted regression trees were 77.5% ± 4.6% (25100 trees) and 79.7–94.9%, respectively. The black bars represent the resampled boosted regression trees for the same variables. Predicted fertility is expressed as a function of variation in (**b**) infant mortality, (**c**) household size, (**d**) access to any form of contraception, (**e**) family-planning quality, (**f**) female education, (**g**) proportion of Catholics or Muslims, and (**h**) visits by a community health worker to discuss family-planning and maternal and child health.

of both men and women, yet male contraceptive use is decreasing globally [29]. Potential reasons could be local views on gender equality, respect and dignity for women, and the proportion of women participating in the workforce [50]. Child marriage and gender-based violence are positively correlated with low contraceptive use and increased fertility in some conditions [51,52]; however, women who experience child-marriage have a higher modern contraceptive use (e.g., female and male sterilisation, intra-uterine device, oral hormonal pills, vaginal barrier methods) compared to adult married women [53]. Our findings also suggest that increasing the availability of contraceptives can be effective in reducing fertility where parents do not have access to secondary education. Fundamentally, contraceptive use is closely linked to infant mortality. Allowing citizens to choose family planning by providing readily available, modern methods of contraception, could improve infant survival because parents can plan and space their births, thereby investing in higher-quality care and increased available resources for each child. This could also improve maternal and child-health outcomes globally.

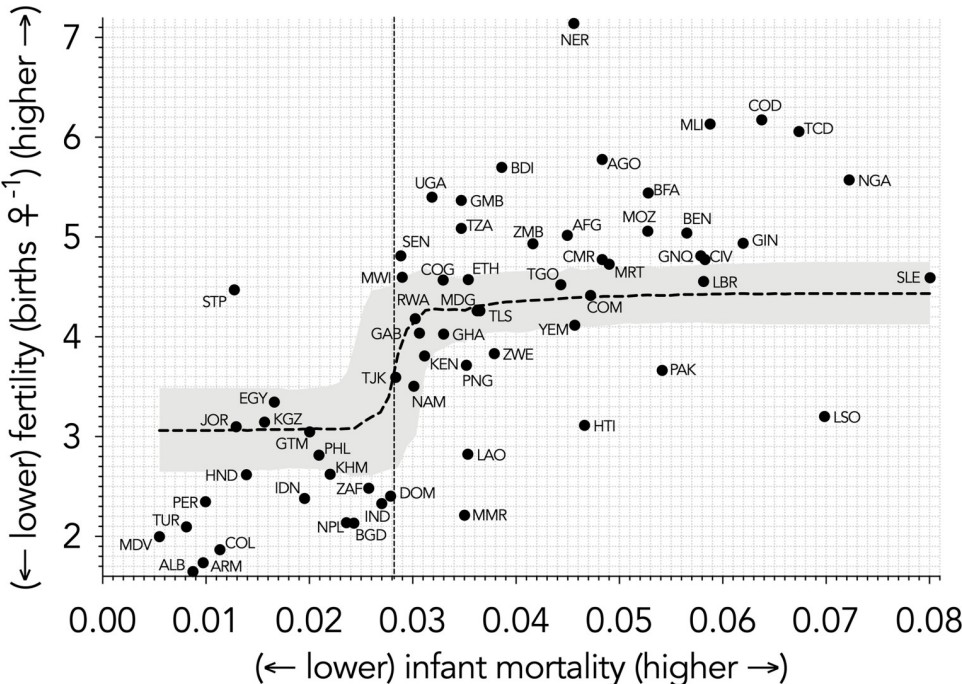

**Fig 3. Predicted fertility as a function of variation in infant mortality (raw data from 64 countries for which both variables were available superimposed onto boosted regression tree relationship).** Once infant mortality has exceeded a threshold of approximately 0.0282 (28.2/1000 live births), fertility increases precipitously. Country codes provided in Fig 1.

**Table 3. General least-squares models of the highest-ranked variable from each theme (*i.* family-planning availability, *ii.* family-planning quality, *iii.* education, *iv.* religion, *v.* mortality, *vi.* socio-economics) in relation to variation in fertility among 64 low- and middle-income countries.**

| model | BICᵃ | *w*BICᵇ | psR²mcfᶜ | psR²csᵈ | psR²cuᵉ |
|---|---|---|---|---|---|
| FEᶠ+HSᵍ | 129.40 | 0.687 | 0.33 | 0.55 | 0.61 |
| FE+ACCʰ | 131.24 | 0.273 | 0.32 | 0.54 | 0.59 |
| IMⁱ+HS | 136.90 | 0.016 | 0.28 | 0.50 | 0.54 |
| *ALL*ʲ | 137.45 | 0.012 | 0.28 | 0.49 | 0.54 |
| FE+IM | 139.72 | 0.004 | 0.24 | 0.44 | 0.48 |
| CHWᵏ+HS | 140.21 | 0.003 | 0.26 | 0.47 | 0.51 |
| FE | 139.99 | 0.003 | 0.24 | 0.44 | 0.48 |
| CHW+FE | 179.18 | <0.001 | 0.14 | 0.30 | 0.32 |
| CHW+CMˡ | 168.69 | <0.001 | 0.08 | 0.17 | 0.19 |
| CHW+IM | 161.52 | <0.001 | 0.12 | 0.26 | 0.28 |
| CHW+ACC | 146.90 | <0.001 | 0.22 | 0.41 | 0.45 |
| CHW+QUAᵐ | 170.65 | <0.001 | 0.07 | 0.15 | 0.16 |
| FE+QUA | 196.68 | <0.001 | -0.10 | -0.28 | -0.31 |
| CM+IM | 196.16 | <0.001 | -0.10 | -0.27 | -0.30 |
| CM+ACC | 156.57 | <0.001 | 0.16 | 0.31 | 0.35 |
| CM+QUA | 149.35 | <0.001 | 0.20 | 0.39 | 0.43 |
| CM+HS | 234.77 | <0.001 | -0.35 | -1.33 | -1.45 |
| IM+ACC | 144.08 | <0.001 | 0.24 | 0.44 | 0.48 |
| IM+QUA | 147.39 | <0.001 | 0.21 | 0.41 | 0.45 |

*(Continued)*

**Table 3.** (Continued)

| model | BIC[a] | wBIC[b] | psR$^2$mcf[c] | psR$^2$cs[d] | psR$^2$cu[e] |
|---|---|---|---|---|---|
| CHW | 162.52 | <0.001 | 0.12 | 0.25 | 0.27 |
| CM | 168.85 | <0.001 | 0.05 | 0.11 | 0.12 |
| IM | 171.01 | <0.001 | 0.04 | 0.08 | 0.09 |
| ACC | 158.53 | <0.001 | 0.12 | 0.25 | 0.27 |
| QUA | 150.32 | <0.001 | 0.17 | 0.34 | 0.37 |
| HS | 172.12 | <0.001 | 0.03 | 0.07 | 0.07 |
| *intercept-only* | 172.40 | <0.001 | - | - | - |

[a]**BIC** = Bayesian information criterion; [b]**wBIC** = Bayesian information criterion weight (≈ model probability); [c]**psR$^2$mcf** = pseudo-R$^2$ (McFadden metric); [d]**psR$^2$cs** = pseudo-R$^2$ (Cox & Snell metric); [e]**psR$^2$cu** = pseudo-R$^2$ (Craig & Uhler metric); [f]**FE** = female years of completed education; [g]**HS** = mean number of individuals per household; [h]**ACC** = access to any form of contraception; [i]**IM** = infant mortality; [j]**ALL** = saturated model (including all highest-ranked variables); [k]**CHW** = visitation from community health workers; [l]**CM** = % Catholic or Muslim; [m]**QUA** = quality of family planning index.

## Strengths and limitations

A major strength of our study is using nationally representative surveys from the Demographic and Health Surveys and other databases, which suggests general applicability to low- and middle-income nations. Further, applying several different modelling frameworks that confirmed the main contributors added robustness to our findings. Our design meant that possible biases exist, such as potentially hiding spatial variation within countries when relying on national averages. There was also the potential for systematic differences between countries in terms of data collection and reporting (i.e., unstated variation in the number of surveys completed for each country). Other limitations include using only the most recent, complete data, which perhaps masks temporal variation in the underlying relationships, and the potential mismatching of some variables derived from surveys occurring at different times.

In conclusion, efforts to increase infant survival, particularly in low- and middle-income countries, would in turn reduce fertility and improve child and maternal health outcomes. An important component of these activities is to provide both women and men the choice to access non-coercive, quality family-planning services. Overall, there is more that can be done to aid in meeting the initiatives of the Sustainable Development Goals of the United Nations, which, if unmet, will see global increases in fertility, more child deaths, and more birth-related deaths among women.

## Supporting information

**S1 Fig. Regionalization map.** World map showing four regional classes used as a random effect in the general linear mixed-effects models (*n* = 64 countries). Country codes per region (3-letter ISO): **Latin America & Caribbean** (COL, DOM, GTM, HTI, HND, PER); **Sub-Saharan Africa** (AGO, BEN, BFA, BDI, CMR, TCD, COM, COD, COG, CIV, GNQ, ETH, GAB, GMB, GHA, GIN, KEN, LSO, LBR, MDG, MWI, MLI, MRT, MOZ, NAM, NER, NGA, RWA, STP, SEN, SLE, ZAF, TZA, TGO, UGA, ZMB, ZWE); **Middle East & North Africa** (EGY, JOR, YEM); **Europe & Central Asia** (ALB, ARM, KGZ, TJK, TUR); **South Asia** (AFG, BGD, IND, MDV, NPL, PAK), **East Asia & Pacific** (KHM, IDN, LAO, MMR, PNG, PHL, TLS). Map generated in QGIS Version 3.26.1-Buenos Aires using an open-licence layer (https://hub.arcgis.com/datasets/esri::world-countries-generalized/explore?location=-0.684955%2C0.000000%2C2.40).
(EPS)

**S2 Fig. Theme 1 results—availability of family planning.** Boosted regression tree results (variable relative performance) for availability of family planning among 52 low- and middle-income countries (available countries in non-imputed dataset). Cross-validation % deviance explained by the final model of 32750 trees: 52.4 ± 7.0%. **Community health workers** = visitation by a community health worker; **social media marketing** = social marketing of subsidised contraceptives; **community-based distribution** = community-based distribution of family-planning; **access** = 'access' index comprising of indicators for availability of family-planning from the Family Planning Effort Index [1]; **logistics and transport** = logistics and transport; **private sector** = involvement of private-sector agencies and groups.
(TIFF)

**S3 Fig. Theme 2 results—quality of family planning.** Boosted regression tree results (variable relative performance) for quality of family planning among 56 low- and middle-income countries (available countries in non-imputed dataset). Cross-validation % deviance explained by the final model of 28650 trees: 53.4 ± 10.3%. **Any form of contraception** = proportion of a population who have access to any form (modern and/or traditional) contraception; **quality** = 'quality' index of family-planning indicators from the National Composite Index on Family Planning [2]; **no contraception** = proportion of a population who have no access to contraception; **traditional contraception** = proportion of a population who have access to traditional forms of contraception only; **modern contraception** = proportion of a population who have access to modern forms of contraception.
(TIFF)

**S4 Fig. Theme 5 results—mortality.** Boosted regression tree results (variable relative performance) for mortality among 29 low- and middle-income countries (available countries in non-imputed dataset). Cross-validation % deviance explained by the final model of 39800 trees: 62.9 ± 14.7%. **Infant mortality** (deaths per 1000 births); **battle-related mortality** (deaths per capita); **maternal mortality** (deaths per 100,000 live births)
(TIFF)

**S5 Fig. Theme 6 results—socio-economics.** Boosted regression tree results (variable relative performance) for socio-economic indicators among 61 low- and middle-income countries (available countries in non-imputed dataset). Cross-validation % deviance explained by the final model of 39900 trees: 75.3 ± 4.8%. **Three generations** = households with at least three generations residing; **house size** = mean number of household members; **wealth inequality** = bottom 50% of net personal wealth.
(TIFF)

**S6 Fig. Final phase results.** Boosted regression tree results (variable relative performance) of the final phase variables based on multiple-imputed *versus* missing-data datasets for 64 and 51 low- and middle-income countries, respectively. Cross-validation % deviance explained by the final model: 85.4% ± 3.7% (25100 trees) and 82.6% ± 3.4% (27150 trees) for the imputed and missing datasets, respectively. **House size** = mean number of individuals per household; **community health workers** = visitation from community health workers.
(TIFF)

**S1 Table. Variable correlation matrix.** Correlation (Kendall's $\tau$) matrix of the highest-ranked variable(s) from each of the six thematic modelling phases (1. family-planning availability, 2. family-planning quality, 3. education, 4. religion, 5. mortality, 6. socio-economics) among 64 low- and middle-income countries. [a]**comm work** = visitation by a community healthcare

worker; [b]**education** = female years of education; [c]**Cathol+Musl** = % of population Catholic or Muslim; [d]**infant mort** = infant mortality; [e]**access** = access to any form of contraception; [f]**quality** = quality of family planning index.
(DOCX)

**S2 Table. Theme 1 results—availability of family planning.** General linear models for indicators of availability of family-planning in relation to variation in fertility among 52 low- and middle-income countries (available countries in non-imputed dataset). [a]**k** = number of parameters; [b]**LL** = log-likelihood; [c]**ΔBIC** = difference in Bayesian information criterion between model and top-ranked model; [d]**wBIC** = Bayesian information criterion weight (≈ model probability); [e]**%DE** = % deviance explained; [f]**ACC** = 'access' index comprising indicators for availability of family planning from the Family Planning Effort Index [1]; [g]**CBDT** = community-based distribution of family planning; [h]**SOCM** = social marketing of subsidised contraceptives; [i]**CHW** = visitation by a community health worker; [j]**LOGT** = logistics and transport; [k]**PRIV** = involvement of private-sector agencies and groups; [l]**ALL** = saturated model (all variables included).
(DOCX)

**S3 Table. Theme 2 results—quality of family planning.** General linear models for indicators of quality of family-planning in relation to variation in fertility among 56 low- and middle-income countries (available countries in non-imputed dataset). [a]**k** = number of parameters; [b]**LL** = log-likelihood; [c]**ΔBIC** = difference in Bayesian information criterion between model and top-ranked model; [d]**wBIC** = Bayesian information criterion weight (≈ model probability); [e]**%DE** = % deviance explained; [f]**QUA** = 'quality' index from the National Composite Index of Family Planning [2]; [g]**NC** = access to no form of contraception; [h]**AC** = access to any form of contraception; [i]**MC** = access to modern contraceptives; [k]**TC** = access to traditional contraceptives.
(DOCX)

**S4 Table. Theme 5 results—mortality.** General linear models for indicators of mortality in relation to fertility among 29 low- and middle-income countries (available countries in non-imputed dataset). [a]**k** = number of parameters; [b]**LL** = log-likelihood; [c]**ΔBIC** = difference in Bayesian information criterion between model and top-ranked model; [d]**wBIC** = Bayesian information criterion weight (≈ model probability); [e]**%DE** = % deviance explained; [f]**IM** = infant mortality (deaths per 1000 births); [g]**BRD** = battle-related deaths per capita; [h]**MM** = maternal mortality (deaths per 100,000 live births).
(DOCX)

**S5 Table. Theme 6 results—socio-economics.** General linear models for indicators of socio-economics in relation to fertility among 61 low- and middle-income countries (available countries in non-imputed dataset). [a]**k** = number of parameters; [b]**LL** = log-likelihood; [c]**ΔBIC** = difference in Bayesian information criterion between model and top-ranked model; [d]**wBIC** = Bayesian information criterion weight (≈ model probability); [e]**%DE** = % deviance explained; [f]**GEN** = proportion of household with three generations residing; [g]**HS** = household size (number of members); [h]**WLTH** = bottom 50% of net personal wealth from the World Inequality Database.
(DOCX)

**S1 File. Description of indicators within each index.**
(DOCX)

## Author Contributions

**Conceptualization:** Corey J. A. Bradshaw, Jane Heyworth, Peter N. Le Souëf.

**Data curation:** Corey J. A. Bradshaw, Claire Perry, Melinda A. Judge, Chitra M. Saraswati, Jane Heyworth.

**Formal analysis:** Corey J. A. Bradshaw, Claire Perry, Chitra M. Saraswati.

**Funding acquisition:** Peter N. Le Souëf.

**Methodology:** Corey J. A. Bradshaw, Chitra M. Saraswati, Peter N. Le Souëf.

**Project administration:** Corey J. A. Bradshaw, Melinda A. Judge, Jane Heyworth, Peter N. Le Souëf.

**Supervision:** Corey J. A. Bradshaw, Jane Heyworth, Peter N. Le Souëf.

**Validation:** Corey J. A. Bradshaw.

**Visualization:** Corey J. A. Bradshaw.

**Writing – original draft:** Claire Perry.

**Writing – review & editing:** Corey J. A. Bradshaw, Melinda A. Judge, Chitra M. Saraswati, Jane Heyworth, Peter N. Le Souëf.

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
