## [Decision Letter · Decision Letter 0]

5 Jul 2022

PONE-D-22-06894Lower infant mortality and access to contraception reduce fertility in low- and middle-income nationsPLOS ONE

Dear Dr. Judge,

Thank you for submitting your manuscript to PLOS ONE. After careful consideration, we feel that it has merit but does not fully meet PLOS ONE’s publication criteria as it currently stands. Therefore, we invite you to submit a revised version of the manuscript that addresses the points raised during the review process.

We look forward to receiving your revised manuscript.

Kind regards,

Joshua Amo-Adjei, Ph.D

Academic Editor

PLOS ONE

Journal Requirements:

2. We note that Supplemental material - Figure 1 in your submission contain mapimages which may be copyrighted. All PLOS content is published under the Creative Commons Attribution License (CC BY 4.0), which means that the manuscript, images, and Supporting Information files will be freely available online, and any third party is permitted to access, download, copy, distribute, and use these materials in any way, even commercially, with proper attribution. For these reasons, we cannot publish previously copyrighted maps or satellite images created using proprietary data, such as Google software (Google Maps, Street View, and Earth). For more information, see our copyright guidelines: http://journals.plos.org/plosone/s/licenses-and-copyright.

   a. You may seek permission from the original copyright holder of to Supplemental material - Figure 1 publish the content specifically under the CC BY 4.0 license. 

Reviewers' comments:

Reviewer's Responses to Questions

**Comments to the Author**

1. Is the manuscript technically sound, and do the data support the conclusions?

Reviewer #1: Yes

2. Has the statistical analysis been performed appropriately and rigorously? 

Reviewer #1: Yes

3. Have the authors made all data underlying the findings in their manuscript fully available?

Reviewer #1: Yes

4. Is the manuscript presented in an intelligible fashion and written in standard English?

Reviewer #1: Yes

5. Review Comments to the Author

Reviewer #1: Dear Editor,

Thank you for the opportunity to review the manuscript by Bradshaw entitled “Lower infant mortality and access to contraception reduce fertility in low- and middle-income nations”. The authors obtained publicly available datasets from different sources. I find the study to be novel. The authors also adopted several robust statistical analytical techniques and their main finding is that reduction in infant mortality and improved access to contraception could lead to reduction in global fertility. The authors have acknowledged some of the key limitations in their study. That notwithstanding I have provided some suggestions that could improve the paper. Kindly see below my comments.

Key questions

1. I might be wrong but I am tempted to believe that the manuscript was not intended for Plos One. Per whatever reason, including rejection from a different journal which is more likely to be from BMJ Global Health (see page 19, lines 405-409) the authors could have taken their time to reformat it to suite the submission requirements of this journal. That said Line 89 “Hence, our study suggests that to decrease global fertility, activities that decrease infant mortality and increase access to quality, non-coercive family-planning services will have the greatest effect”. In as much as this is a great suggestion, already available data from the World Bank for instance (see https://data.worldbank.org/indicator/SP.DYN.TFRT.IN?locations=XO) has shown that the average fertility rate for LMICs is about 2.5 (2020) which is even expected to reduce further due to several reasons/interventions including your findings. The bone of contention here is that now some countries are devising several strategies to increase their fertility because it is now below replacement levels (see https://www.euro.who.int/__data/assets/pdf_file/0010/73954/EN63.pdf).So making a general suggestion to decrease global fertility is an argument that will be out of place. This needs to be contextualised.

Introduction

2. Page 6 line, 104, it seems the word ‘services’ is missing

3. Kindly provide references for the few studies “Few studies have investigated the relationship between socio-economic conditions and fertility among nations”(page 6 line 110-112).

4. Although the background is well situated in the SDGs, critical issues are missing

a. The authors have failed to expand the literature review on infant mortality and fertility rate

b. They have also failed to situate the study in a given framework. For example, the proximate determinants model could be used

Methods

5. Well described methods but a minor comment, please provide justification for using only datasets from the DHS between 2010 and 2018. There are countries with relatively new datasets https://dhsprogram.com/data/available-datasets.cfm. I am inferring from the data used http://track20.org/pages/data_analysis/policy/NCIFP.php which has the most recent version to be 2017, if that is the reason then it should be explicitly stated.

6. It will also be great fill the STROBE checklist and add it as a supplementary file.

Results

7. The first part of the results could be moved to the methods section (page 11, line 230-241)

Discussion

8. To a greater extent the authors have discussed the key findings. If the authors take on board my suggestion on the need to incorporate a framework in the study, then this should also reflect in their discussion.

9. It will be great to create a sub-heading to discuss the strength and limitations of the study in detail. Some of the limitations worth discussing are: the timeframe for the study, the different survey years for some of the DHSs, the cross-sectional nature of some of the datasets etc.

10. My main concern with the framing of the paper is “reduction in global fertility”. That is true for most low-and middle-income countries but this is not the case in every country. Some countries are even struggling with replacement levels and this should also be discussed in detail.

11. The conclusions and policy implications are not explicitly discussed. This should be the thrust of the paper. A sub-heading/section should be created to discuss this also.

6. PLOS authors have the option to publish the peer review history of their article (what does this mean?). If published, this will include your full peer review and any attached files.

Reviewer #1: No

---

## [Author Response · Author response to Decision Letter 0]

6 Sep 2022

Response to reviewers

Reviewer #1

1. I might be wrong but I am tempted to believe that the manuscript was not intended for Plos One. Per whatever reason, including rejection from a different journal which is more likely to be from BMJ Global Health (see page 19, lines 405-409) the authors could have taken their time to reformat it to suite the submission requirements of this journal. 

RESPONSE #1: We have now formatted for PLOS ONE. We acknowledge our oversight. 

That said Line 89 “Hence, our study suggests that to decrease global fertility, activities that decrease infant mortality and increase access to quality, non-coercive family-planning services will have the greatest effect”. In as much as this is a great suggestion, already available data from the World Bank for instance (data.worldbank.org/indicator/SP.DYN.TFRT.IN?locations=XO) has shown that the average fertility rate for LMICs is about 2.5 (2020) which is even expected to reduce further due to several reasons/interventions including your findings. The bone of contention here is that now some countries are devising several strategies to increase their fertility because it is now below replacement levels (see euro.who.int/__data/assets/pdf_file/0010/73954/EN63.pdf). So making a general suggestion to decrease global fertility is an argument that will be out of place. This needs to be contextualised.

RESPONSE #2: We respectfully disagree. There is little chance of global population decline this century. For example, the United Nations Population Division’s latest cohort-component models projects a median scenario of reaching 10.4 billion by the 2080s, and maintaining approximately that size until 2100 (assuming fertility rates continue declining on average).

The contention is therefore based on the observation a few high-income nations are bucking the global trend by experiencing slight population declines and changing age structures, and the misplaced notion that this represents a problem. 

But to slow and perhaps even reverse climate change [1], as well as mitigate the extinction crisis underway [2], we are obliged to reduce consumption globally [3, 4] all this within an economic system that is essentially ‘broken’ [5]. Shrinking human populations will contribute to that goal (provided we simultaneously reduce per-capita consumption). The oft-touted ‘crisis’ of ageing populations is founded on the erroneous notion that it will lead to economic catastrophes for the affected countries. Indeed, countries like South Korea and Japan have declining populations [6], others like Italy are stable and will be declining soon [7], and some countries like Australia are only growing because of net immigration [8].

The reason for exaggeration of a ‘crisis’ generally comes down to the overly simplistic ‘dependency ratio’ [9], which has several different forms but generally compares the number of people in the labour force against those who have retired from it. The idea here is that once the number of people no longer in the labour force exceeds the number of those in the labour force, the latter can no longer support the entirety of the former. 

This simplistic 1:1 relationship essentially assumes that you need one person working to support one retired person. But this is a flawed interpretation for several reasons. First, in any country experiencing population decline (i.e., mainly high-income nations), there is almost always a form of national superannuation (retirement savings). This means that labourers save money in a special investment fund (usually guaranteed or supported by government co-contributions) such that by the time of retirement, there is sufficient funds to persist until the end of life. Certainly, some superannuation schemes are better than others, but the idea that the working support the non-working is not only simplistic, it is mostly wrong. 

The dependency ratio also assumes that anyone too young to be in the labour force is irrelevant for a nation’s economy. But this too is incorrect. A declining population also has a changing age structure, meaning that there are fewer young people (children). With fewer children comes fewer societal expenses in many aspects (e.g., education, transport, housing, etc.). It turns out that once you include children, dependency ratios do not change as much as those including only adults [8, 10].

The argument tends to focus instead on the increasing ratio of retired people requiring increased medical spending. This too is an oversimplification. People are today living longer and have more years of healthy life than they have ever before [9], and public healthcare is a self-sustaining concept given that support for healthy people in their younger years reduces the time spent unhealthy later. Dependency ratios also assume a static set of conditions between labourers and retirees. But this is so simplistic as to be patently ridiculous. No longer do most people retire at the age of 55 and cease any meaningful contribution to the economy. Forgetting unpaid volunteer work for the moment (which is a sizeable, yet non-valued aspect of most economies) [11], people are working much later in life, have flexible work arrangements (COVID has emphasised this), and are generally contributing to economies well into their retirement years. Assuming fixed conditions is an ageist concept — it essentially treats retirees as useless members of society.

Simplistic dependency ratios used to justify a looming demographic ‘crisis’ are also inherently xenophobic and racist. Because the Earth’s human population is nowhere near reaching a peak or decline, there is a plentiful pool of able-bodied people of working age in most of the world. The problem of insufficient number of labourers in any one country is then entirely based on a distribution issue — limited or suffocating immigration policies (including welcoming and open refugee policies) could ‘fix’ any labour shortages anywhere with the right policies. There is ample evidence now that migrants provide net benefits to the receiving economies [12], not the other way around.

Invoking the idea of a demographic crises also ignores the overwhelming benefits population reductions have for the average person (mitigation of climate change and biodiversity loss notwithstanding). Fewer people clambering for insufficient housing means that lifestyles improve and become more affordable. Having fewer people alleviates potential food-supply and -distribution problems. Having fewer people means fewer cars on the streets, easier access to public transportation, more affordable medical services, and less-competitive educational opportunities. For the average person, fewer people = better life.

Frankly, politicians (and their corporate backers) echo the trope that population declines are bad mainly because fewer consumers mean lower net profits for shareholders. The mega-rich will be slightly less mega-rich if there is a moderate drop in total number of consumers. Corporate capture of governments worldwide perpetuates the myth that an ageing population is bad for us, when it is in fact the opposite.

As such, we are confident in our wording as expressed.

2. Page 6 line, 104, it seems the word ‘services’ is missing

RESPONSE #3: Corrected.

3. Kindly provide references for the few studies “Few studies have investigated the relationship between socio-economic conditions and fertility among nations”(page 6 line 110-112).

RESPONSE #4: We have added Janowitz [13] and Bollen et al. [14].

4. Although the background is well situated in the SDGs, critical issues are missing

a. The authors have failed to expand the literature review on infant mortality and fertility rate

RESPONSE #5: We chose to pose hypotheses based on evidence from existing research about the link between fertility and infant mortality.

However, we have elected to expand the justification and references in the Introduction with the following:

“Infant mortality has been linked to increased fertility due to the ‘insurance’ or ‘replacement’ effect [15, 16], and child morbidity and mortality are themselves exacerbated by higher household sizes (i.e., number of individuals per household) [17-23].”

b. They have also failed to situate the study in a given framework. For example, the proximate determinants model could be used

RESPONSE #6: As we stated in Response #5, we have updated our hypothesis with appropriate citations about the well-established link. While Bongaarts {Bongaarts, 1978 #17209}was examining family-level determinants and individual choices made within certain socio-economic contexts, we acknowledge that citing Bongaarts study is wise, so we have included it in the new text highlighted in Response #5.

5. Well described methods but a minor comment, please provide justification for using only datasets from the DHS between 2010 and 2018. There are countries with relatively new datasets https://dhsprogram.com/data/available-datasets.cfm. I am inferring from the data used http://track20.org/pages/data_analysis/policy/NCIFP.php which has the most recent version to be 2017, if that is the reason then it should be explicitly stated.

RESPONSE #7: Most of the time we have taken to revise the manuscript has been dedicated to overhaul the dataset almost completely by:

i. obtaining the most complete, up-to-date data

ii. improving the data in certain modelling phases by obtaining better indicators of particular variables,

iii. adding more countries to the sample, and

iv. providing a multiple imputation procedure to increase the sample size of countries by robustly imputing some missing data.

We outline these changes briefly here.

• Total fertility is now the mean from 2010 to 2020 (from the World Bank) [24];

• We replaced school completion data with the mean years of education of women aged 15–19 from the Demographic and Health Surveys [25];

• We combined Catholic and Muslim adherents data per country from the Association of Religion Data Archives [26] (previously, we only include Catholics, despite evidence of higher fertility in Muslim-majority nations in Sub-Saharan Africa) [27];

• We now use the newest estimates of infant mortality (2020) from the World Health Organization [28];

• We replaced the poverty variables with the net personal wealth of the bottom 50% from the World Inequality Database [29] as a better measure of mean poverty (including debt);

• We replaced conflict-related deaths with battle-related deaths (World Bank) [24];

• We updated the percentage of household with three generations residing, mean number of household members, and all the family-planning availability variables (access) from the Demographic and Health Surveys Program [25] and the Multiple Indicator Cluster Surveys [30].

• The new sample of countries in the raw dataset is now 64, yet there were still some countries missing data (up to 19% in some variables). We therefore applied multiple imputation by chained equations to maintain the full sample of 64 countries to improve our strength of inference and robustness of results.

• We re-ran all models (general linear/mixed-effects, general least-squares, boosted regression trees) with these new data. Our conclusions hold, yet we did find a few small differences (e.g., the inclusion of household size as an important explanatory variable, and the counter-intuitive positive relationship between family-planning quality and fertility – discussed in the main text).

6. It will also be great fill the STROBE checklist and add it as a supplementary file.

RESPONSE #8: We have not included a STROBE checklist because our study did not directly involve human participants. We did not do any trials, did no patient interventions, and required no ethics approval to access publicly available data.

7. The first part of the results could be moved to the methods section (page 11, line 230-241)

RESPONSE #9: We respectfully disagree. These are important results to help interpret the subsequent modelling outputs.

8. To a greater extent the authors have discussed the key findings. If the authors take on board my suggestion on the need to incorporate a framework in the study, then this should also reflect in their discussion.

RESPONSE #10: Please see Response #6. There is a major disconnect between the family/individual-level proximate determinants model with our national-scale investigation. We argue that we have provided an expansive justification for posing the hypothesis and interpreting the results, especially in the revision with a broader coverage of the literature.

9. It will be great to create a sub-heading to discuss the strength and limitations of the study in detail. Some of the limitations worth discussing are: the timeframe for the study, the different survey years for some of the DHSs, the cross-sectional nature of some of the datasets etc.

RESPONSE #10: We have now added a ‘Strengths and limitations’ section as recommended.

10. My main concern with the framing of the paper is “reduction in global fertility”. That is true for most low-and middle-income countries but this is not the case in every country. Some countries are even struggling with replacement levels and this should also be discussed in detail.

RESPONSE #11: We have addressed this comment at length in Response #2.

11. The conclusions and policy implications are not explicitly discussed. This should be the thrust of the paper. A sub-heading/section should be created to discuss this also.

RESPONSE #12: We contend that while our results have strong policy implications that we discuss at the end of the paper, our manuscript does not analyse policies per se. Furthermore, policies are generally rolled out at subnational scales, and so require bespoke, context-specific language and legislation to be effective. We cannot possibly prescribe such an array of national and sub-national bespoke policies, even if we were policy experts. However, our results do broadly support the notion that improving family planning and reducing child mortality will have some of the greatest benefits to low- and middle-income nations in the future, regardless of what policies are invoked to achieve those broad aims.

References

1. IPCC. AR6 Climate Change 2021. UK: Cambridge Univ Press, 2021.

2. Bradshaw CJA, Ehrlich PR, Beattie A, Ceballos G, Crist E, Diamond J, et al. Underestimating the challenges of avoiding a ghastly future. Front Conserv Sci. 2021;1:9. doi: 10.3389/fcosc.2020.615419.

3. Wackernagel M, Lin D, Evans M, Hanscom L, Raven P. Defying the footprint oracle: implications of country resource trends. Sustainability. 2019;11(7):2164-. doi: 10.3390/su11072164.

4. Wackernagel M, Hanscom L, Jayasinghe P, Lin D, Murthy A, Neill E, et al. The importance of resource security for poverty eradication. Nat Sustain. 2021;4:731-8. doi: 10.1038/s41893-021-00708-4.

5. Dasgupta P. The Economics of Biodiversity: The Dasgupta Review. London: HM Treasury, 2021.

6. ourworldindata.org/grapher/future-population-projections-by-country?country=BRA~USA~JPN~ITA~KOR.

7. Raj A, Ghule M, Battala M, Dasgupta A, Ritter J, Nair S, et al. Brief report: parent–adolescent child concordance in social norms related to gender equity in marriage – findings from rural India. J Adolesc. 2014;37(7):1181-4. doi: 10.1016/j.adolescence.2014.08.006.

8. Bradshaw CJA, Brook BW. Implications of Australia's population policy for future greenhouse-gas emissions targets. Asia Pac Pol Stud. 2016;3:249-65. doi: 10.1002/app5.135.

9. ourworldindata.org/age-structure.

10. Bradshaw CJA, Brook BW. Human population reduction is not a quick fix for environmental problems. Proc Natl Acad Sci USA. 2014;111(46):16610-5. doi: 10.1073/pnas.1410465111.

11. probonoeconomics.com/News/the-economic-value-of-volunteering.

12. brookings.edu/blog/up-front/2018/06/19/refugees-are-a-win-win-win-formula-for-economic-development.

13. Janowitz BS. An empirical study of the effects of socioeconomic development on fertility rates. Demography. 1971;8(3):319-30. doi: 10.2307/2060620.

14. Bollen KA, Glanville JL, Stecklov G. Socioeconomic Status and Class in Studies of Fertility and Health in Developing Countries. Annual Review of Sociology. 2001;27:153-85.

15. van Soest A, Saha UR. Relationships between infant mortality, birth spacing and fertility in Matlab, Bangladesh. PLoS One. 2018;13(4):e0195940. doi: 10.1371/journal.pone.0195940.

16. Bongaarts J. A framework for analyzing the proximate determinants of fertility. Pop Dev Rev. 1978;4(1):105-32. doi: 10.2307/1972149.

17. Bradshaw CJA, Otto SP, Mehrabi Z, Annamalay AA, Heft-Neal S, Wagner Z, et al. Testing the socioeconomic and environmental determinants of better child-health outcomes in Africa: a cross-sectional study among nations. BMJ Open. 2019;9(9):e029968. doi: 10.1136/bmjopen-2019-029968.

18. Annim SK, Awusabo-Asare K, Amo-Adjei J. Household nucleation, dependency and child health outcomes in Ghana. J Biosoc Sci. 2015;47(5):565-92. doi: 10.1017/S0021932014000340.

19. Root G. Population density and spatial differentials in child mortality in Zimbabwe. Soc Sci Med. 1997;44(3):413-21. doi: 10.1016/S0277-9536(96)00162-1.

20. Anand A, Roy N. Transitioning toward Sustainable Development Goals: the role of household environment in influencing child health in Sub-Saharan Africa and South Asia using recent demographic health surveys. Front Pub Health. 2016;4:87. doi: 10.3389/fpubh.2016.00087.

21. Wolfe BL, Behrman JR. Determinants of child mortality, health, and nutrition in a developing country. J Dev Econ. 1982;11(2):163-93. doi: 10.1016/0304-3878(82)90002-5.

22. Lolekha S, Tanthiphabha W, Sornchai P, Kosuwan P, Sutra S, Warachit B, et al. Effect of climatic factors and population density on varicella zoster virus epidemiology within a tropical country. Am J Trop Med Hyg. 2001;64(3):131-6. doi: 10.4269/ajtmh.2001.64.131.

23. Greiner KA, Li C, Kawachi I, Hunt DC, Ahluwalia JS. The relationships of social participation and community ratings to health and health behaviors in areas with high and low population density. Soc Sci Med. 2004;59(11):2303-12. doi: 10.1016/j.socscimed.2004.03.023.

24. data.worldbank.org.

25. dhsprogram.com.

26. thearda.com.

27. Westoff CF, Bietsch K. Religion and Reproductive Behavior in Sub-Saharan Africa. DHS Analytical Studies No. 48. Rockville, Maryland, USA: ICF International, 2015.

28. who.int/data/gho.

29. wid.world/data.

30. mics.unicef.org.

---

## [Decision Letter · Decision Letter 1]

26 Dec 2022

Lower infant mortality, higher household size, and more access to contraception reduce fertility in low- and middle-income nations

PONE-D-22-06894R1

Dear Dr. Judge,

We’re pleased to inform you that your manuscript has been judged scientifically suitable for publication and will be formally accepted for publication once it meets all outstanding technical requirements.

Kind regards,

Joshua Amo-Adjei, Ph.D

Academic Editor

PLOS ONE

Additional Editor Comments (optional):

Reviewers' comments:

Reviewer's Responses to Questions

**Comments to the Author**

1. If the authors have adequately addressed your comments raised in a previous round of review and you feel that this manuscript is now acceptable for publication, you may indicate that here to bypass the “Comments to the Author” section, enter your conflict of interest statement in the “Confidential to Editor” section, and submit your "Accept" recommendation.

Reviewer #1: All comments have been addressed

2. Is the manuscript technically sound, and do the data support the conclusions?

Reviewer #1: Yes

3. Has the statistical analysis been performed appropriately and rigorously? 

Reviewer #1: Yes

4. Have the authors made all data underlying the findings in their manuscript fully available?

Reviewer #1: Yes

5. Is the manuscript presented in an intelligible fashion and written in standard English?

Reviewer #1: Yes

6. Review Comments to the Author

Reviewer #1: The authors have satisfactorily addressed all the major issues I raised in the previous version of the manuscript. Subject to the Editor's decision, the manuscript is potentially acceptable for publication at this stage.

7. PLOS authors have the option to publish the peer review history of their article (what does this mean?). If published, this will include your full peer review and any attached files.

Reviewer #1: **Yes: **Abdul-Aziz Seidu

---

## [Editor Report · Acceptance letter]

30 Jan 2023

PONE-D-22-06894R1 

Lower infant mortality, higher household size, and more access to contraception reduce fertility in low- and middle-income nations 

Dear Dr. Judge:

I'm pleased to inform you that your manuscript has been deemed suitable for publication in PLOS ONE. Congratulations! Your manuscript is now with our production department. 

Kind regards, 

on behalf of

Dr. Joshua Amo-Adjei 

Academic Editor

PLOS ONE